# Application of Augmented Reality Technology as a Dietary Monitoring and Control Measure Among Adults: A Systematic Review

**DOI:** 10.3390/nu17243893

**Published:** 2025-12-12

**Authors:** Gabrielle Victoria Gonzalez, Bingjing Mao, Ruxin Wang, Wen Liu, Chen Wang, Tung Sung Tseng

**Affiliations:** 1Community Health Sciences and Policy Program, School of Public Health, Louisiana State University Health Sciences Center, New Orleans, LA 70112, USA; ggonz4@lsuhsc.edu (G.V.G.); bmao@lsuhsc.edu (B.M.); 2Computer Science, College of AI, Cyber and Computing, University of Texas at San Antonio, San Antonio, TX 78249, USA; ruxin.wang@utsa.edu; 3Computer Science, Lyle School of Engineering, Southern Methodist University, Dallas, TX 75205, USA; wenliu@mail.smu.edu (W.L.); cwang6@mail.smu.edu (C.W.)

**Keywords:** augmented reality, dietary monitoring, dietary control, usability, nutrition technology

## Abstract

**Background/Objectives**: Traditional dietary monitoring methods such as 24 h recalls rely on self-report, leading to recall bias and underreporting. Similarly, dietary control approaches, including portion control and calorie restriction, depend on user accuracy and consistency. Augmented reality (AR) offers a promising alternative for improving dietary monitoring and control by enhancing engagement, feedback accuracy, and user learning. This systematic review aimed to examine how AR technologies are implemented to support dietary monitoring and control and to evaluate their usability and effectiveness among adults. **Methods**: A systematic search of PubMed, CINAHL, and Embase identified studies published between 2000 and 2025 that evaluated augmented reality for dietary monitoring and control among adults. Eligible studies included peer-reviewed and gray literature in English. Data extraction focused on study design, AR system type, usability, and effectiveness outcomes. Risk of bias was assessed using the Cochrane RoB 2 tool for randomized controlled trials and ROBINS-I for non-randomized studies. **Results**: Thirteen studies met inclusion criteria. Since the evidence based was heterogeneous in design, outcomes, and measurement, findings were synthesized qualitatively rather than pooled. Most studies utilized smartphone-based AR systems for portion size estimation, nutrition education, and behavior modification. Usability and satisfaction varied by study: One study found that 80% of participants (N = 15) were satisfied or extremely satisfied with the AR tool. Another reported that 100% of users (N = 26) rated the app easy to use, and a separate study observed a 72.5% agreement rate on ease of use among participants (N = 40). Several studies also examined portion size estimation, with one reporting a 12.2% improvement in estimation accuracy and another showing −6% estimation, though a 12.7% overestimation in energy intake persisted. Additional outcomes related to behavior, dietary knowledge, and physiological or psychological effects were also identified across the review. Common limitations included difficulty aligning markers, overestimation of amorphous foods, and short intervention durations. Despite these promising findings, the existing evidence is limited by small sample sizes, heterogeneity in intervention and device design, short study durations, and variability in usability and accuracy measures. The limitations of this review warrant cautious interpretation of findings. **Conclusions**: AR technologies show promise for improving dietary monitoring and control by enhancing accuracy, engagement, and behavior change. Future research should focus on longitudinal designs, diverse populations, and integration with multimodal sensors and artificial intelligence.

## 1. Introduction

Dietary monitoring, the process of tracking and recording an individual’s food intake to assess nutritional status and its connection to health, is closely related to dietary control, also referred to as dietary or caloric restriction, which involves managing nutritional intake to meet specific health needs [1,2]. Both dietary monitoring and dietary control are methods to improve the nutritional health of adult populations [3,4,5]. Traditional dietary monitoring methods, such as 24 h recalls, food frequency questionnaires (FFQs), and dietary records, rely heavily on self-report, which introduces limitations including recall bias, underreporting, and inconsistent adherence [1,2,6]. Similarly, dietary control measures often depend on self-monitoring strategies like portion control, calorie restriction, and dietary restraint [7]. However, because these methods depend heavily on individuals’ accuracy and consistency, they can lead to misreporting and can promote unhealthy eating behaviors or disordered eating patterns [8]. Ultimately, these limitations reduce the precision and reliability of dietary assessments and may limit the effectiveness of dietary control interventions in real-world settings.

Recent technological advancements, particularly augmented reality (AR), offer promising alternatives and solutions to the limitations of traditional dietary monitoring and control tools. We have already seen success in AR based technology is being implemented in successful healthy behavior change interventions. For example, Odenigbo and colleagues conducted a review of AR and VR technologies in enhancing healthy behavior interventions. AR technology differs from virtual technology (VR) by superimposing digital information into the real-world in real time, allowing users to interact with virtual overlays. Through these immersive features, can visualize food portions, nutritional content, and dietary guidelines, enhancing users’ awareness of their eating behaviors. For example, Naritomi and Yanai created “CalorieCaptorGlass,” which estimates caloric content using image recognition through HoloLens AR glasses [9]. Several AR-based tools also provide real-time feedback, enabling immediate nutritional guidance that supports improved dietary knowledge [10], healthy eating behaviors [3], and reporting of food intake [11].

Beyond its technical advantages, AR can also enhance dietary control through psychological and behavioral mechanisms [12]. The novelty and interactivity of AR can increase user engagement compared to traditional dietary logging [13,14], which many individuals find tedious or time-consuming. By transforming dietary tracking into a more immersive and rewarding experience, AR encourages consistent participation and reduces the cognitive burden associated with food logging [3]. Over time, as AR systems become more accurate and responsive, users can internalize feedback, learn self-correct portion estimations, and strengthen self-regulation skills, which are key components of sustainable dietary control [3,5].

In recent years, AR technology has been increasingly applied in nutritional interventions aiming to monitor diets and promote healthier eating behaviors among adults. Studies have explored its use in portion size estimation, food recognition, and behavior change [14]. The integration of AR with mobile devices, smart glasses, and tablets has expanded its accessibility, while the incorporation of image recognition, fiducial markers, and AI-driven models has enhanced its precision [14,15]. Despite these advances, the diversity of AR systems and the absence of standardized evaluation frameworks present challenges for comparing outcomes across studies [15]. This systematic review aims to synthesize current evidence on how AR technologies are used for dietary monitoring and control, identify gaps in methodological standardization, and evaluate their potential to enhance dietary self-regulation and behavior change.

### 1.1. Rationale

This systematic review assessed the effectiveness of AR technology not only as a tool for dietary monitoring but also as an enhanced method for dietary control that improves nutritional knowledge and food estimation compared to traditional methods. Currently, there is a lack of a comprehensive systematic review of the available research conducted using augmented reality to assess dietary monitoring and control in the literature, particularly how AR tools can improve users’ ability to identify, interpret, and respond to dietary information [16].

Augmented reality research has expanded rapidly in recent years, with research dedicated to understanding how augmented reality technology can be implemented in nutritional studies and interventions. Prior systematic reviews have focused on using AR in food labeling, taste perception, and nutrition and diet education. for food and nutrition [3,4,14]. Nutrition and diet education reviews focused on the advantages and disadvantages of AR and VR for nutritional and diet-related education, as well as the application [3]. These reviews had a common goal to evaluate the effectiveness of AR technology in various nutritional-related interventions. Although these reviews provide important overviews of AR’s potential benefits and limitations, they do not synthesize three domains that are essential to evaluating AR for dietary assessment: (1) usability measured with standardized instruments, such as the System Usability Scale (SUS) or the Computer System Usability Questionnaire (CSUQ); (2) accuracy outcomes, including portion size estimation performance; and (3) behavioral effects relevant to dietary control, such as changes in eating awareness, confidence in dietary knowledge, or reductions in self-report bias. Importantly, dietary control extends beyond dietary monitoring by focusing on how AR-mediated feedback, visualization, or interactivity may influence users’ eating behaviors, decision-making, or regulation of intake. By integrating evidence across these domains, this review provides a focused synthesis not addressed in the previous literature and clarifies AR’s emerging role in supporting both dietary monitoring and dietary control strategies.

For the purposes of this review, several key terms require clarification. Usability refers to the extent to which an AR system can be used effectively, efficiently, and satisfactorily by the intended users. Feasibility reflects the practicality of implementing the AR system within a specific context, including user burden, time demands, and technical reliability. Acceptability describes users’ perceived appropriateness and overall approval of the AR tool based on their cognitive and emotional responses.

This review also distinguishes between two domains of effectiveness. Monitoring effectiveness refers to the degree to which AR enhances the accuracy and precision of dietary monitoring tasks, such as portion size estimation, food identification, or nutrient quantification. In contrast, control effectiveness refers to the influence of AR on behavioral or cognitive outcomes related to dietary control, including improved dietary knowledge, healthier food choices, increased self-regulation, or changes in physiological or psychological responses during eating.

These definitions guided the synthesis and categorization of findings throughout the review and provide a framework for interpreting how AR technologies may support different aspects of dietary behavior.

### 1.2. Objectives

This systematic review examines the application of AR technology as a method to monitor and control dietary behavior in adult populations. Specifically, dietary monitoring will be measured as a way of tracking dietary intake to understand eating patterns, and dietary control will involve actively choosing what and how much food is consumed.
The primary objective of this review is to examine the usability of augmented reality technology.The secondary objective is to examine the effectiveness of augmented reality technology being applied as a dietary monitoring and control method.

### 1.3. Research Question

How is augmented reality (AR) technology, across various devices (apps, glasses, tablets, etc.), implemented to support dietary monitoring and control, and what factors influence its usability among users?

## 2. Materials and Methods

### 2.1. Literature Search

Relative literature for uses of augmented reality technology as a dietary monitoring and control method among adults were identified through an electronic search for papers published from 2000 to 2025. This timeframe was selected to align with the introduction of augmented reality applications beginning in 2000, when ARToolKit was first made available. The review has been registered to PROSPERO and approved by the team. Here is the PROSPERO ID: CRD420251247625 (date 5 December 2025).

Databases searched included PubMed, CINAHL, and Embase, and the initial search was conducted in May 2023. In line with our focus on public health and nutrition applications of AR, we selected PubMed, CINAHL, and Embase since preliminary scoping searches indicated that Scopus and Web of Science largely yielded engineering-centered AR publications without relevance to dietary monitoring or behavioral outcomes. Keywords such as “augmented reality”, “mixed reality”, “dietary control”, “dietary assessment”, “dietary behavior” were used. Full search string terms are detailed in Appendix A.

Study Eligibility Criteria and Search Strategy:

Inclusion criteria for this review were (1) examine augmented reality technology, (2) discuss usability measures of AR technology (3) limit to adults (aged 18 years and older), and (4) English version of full-text peer-reviewed publications, conference papers, and gray literature. Dissertations were not included in this review. The inclusion of gray literature allowed for search of evidence-based materials, such as conference papers, not published in scholarly journals. Additionally, this review was not limited to articles published in the United States.

### 2.2. Data Collection and Selection Process

One independent researcher conducted the data collection and selection process. Although one reviewer conducted the initial title, abstract, and full-text screening, a second reviewer independently verified all screening decisions. Discrepancies were resolved through consensus. Data extraction was conducted by one reviewer and subsequently checked by two additional reviewers, including subject-matter experts in computer science and augmented reality technology, to ensure accuracy of technical descriptions and outcome analyses. LSUHSC library services provided access to articles.

Study Selection:

The initial search yielded 108 relevant records and 5 gray literature sources. After the removal of duplicates (n = 59), articles (n = 49) were screened by title and abstract. Articles not meeting the inclusion criteria (n = 22) were excluded. A full text assessment was conducted on the remaining articles (n = 27). Articles were removed for not discussing augmented reality (n = 9), not discussing dietary monitoring or control (n = 4), not offering free full text articles (n = 2), and narrative reviews (n = 4). The final articles from the initial search and addition of relevant articles from the gray literature included thirteen final articles. The final articles (n = 13) were included in the systematic review [5,17,18,19,20,21,22,23,24,25,26,27,28]. The flowchart of the screening process is presented in Figure 1.

### 2.3. Outcome Definitions

The categorization presented in Table 1 reflects the methodological diversity of the studies included in this review and aligns with common classifications used in behavioral science and public health research. Experimental designs refer broadly to studies in which an intervention is deliberately manipulated to examine its effects on an outcome. Randomized controlled trials fall within this category as a specific subtype distinguished by the random assignment of participants to conditions. Quasi-experimental studies also involve the manipulation of an intervention but lack randomization, which may introduce selection or confounding bias. Cross-sectional survey designs were included to represent studies that collected data at a single time point without manipulating an intervention. Mixed methods studies are presented as a separate category because, although they might include experimental or observational factors, their defining characteristic is the inclusion of both quantitative and qualitative approaches to enhance the interpretation of AR usability or effectiveness. The taxonomy allowed for a summary of the full range of methodological approaches used in the current AR dietary monitoring and control literature. 

### 2.4. Risk of Bias

The Cochrane Risk of Bias tool V2 [RoB2] [29] was used to assess the risk of bias of all included studies that are randomized controlled trials (N = 3) as recommended in the Cochrane Handbook for Systematic Reviews of Interventions [30]. All RCTs were ranked in five different domains, including the randomization process, deviations from intended interventions, missing outcome data, measurement of the outcome, and selection of the reported result. Randomization process includes whether allocation sequence generation and concealment were adequate and whether baseline differences suggested selection bias. Deviations from intended interventions include if participants, personnel, or study procedures deviated from the assigned intervention in a way that could influence outcomes. Missing outcome data includes if attrition, exclusions, or incomplete data could bias results. Measurement of the outcome includes whether outcome assessors were blinded and whether measurement methods were valid and consistently applied. Selection of the reported result determined if reported outcomes were consistent with prespecified plans and free from selective reporting.

Each domain was evaluated using RoB2 signaling questions, which guide assessors through a structured series of yes/no responses to determine whether risk was “low risk”, “some concerns”, or “high risk”. Types of bias include selection bias, performance bias, detection bias, attrition bias, and reporting bias. The overall risk of bias for each RCT was determined by the highest level of concern observed across the five domains. A study was classified as low risk only if all domains were rated low risk; some concerns if at least one domain had some concerns, but none were high risk, and high risk if any domain was rated high risk. Table 2 details the determination of risk per study.

The Cochrane Risk of Bias in non-randomized studies of interventions tool (ROBINS-I) was used to assess the risk of bias of all included studies that are non-randomized studies (N = 10) [31]. All non-randomized studies were ranked in seven domains, including bias due to confounding, bias in the selection of participants into the study, bias in the classification of interventions, bias due to missing data, bias in the measurement of outcomes, and bias in the selection of the reported result.

Bias due to confounding determine whether differences between groups could influence outcomes. Bias in selection of participants into the study questioned if participant recruitment or exclusion could introduce systematic differences. Bias in classification of interventions includes whether intervention status was accurately measured and applied consistently. Bias due to deviations from intended interventions include if deviations from the intended intervention affected outcomes. Bias due to missing data determined if loss to follow-up or incomplete data were related to exposure or outcomes. Bias in measurement of outcomes determines if outcomes measures were valid, reliable, and applied similarly across groups. Lastly, bias in selection of the reported result determined if selective reporting influenced which results were presented.

According to the ROBINS-I guidelines, rankings per domain included low risk, moderate risk, and serious risk. A study with any domain rated serious risk was classified as overall serious risk, studies with only moderate risks were rated moderate, and studies with only low-risk domains were rated overall low risk. Table 3 details the determination of risk per study.

## 3. Results

### 3.1. Characteristics of Studies

In this review, a total of thirteen articles were included to analyze the use of augmented reality technology in dietary monitoring and control. All studies are displayed in Table 4. The studies were selected based on their relevance to the objectives of this review, with a focus on evaluating the effectiveness of a wide range of AR systems used for dietary monitoring and control. The majority of included studies utilized smartphone-based AR applications, while only one incorporated AR glasses. Accordingly, the review examines AR systems broadly rather than focusing specifically on AR glasses. The selected studies, published from 2014 onwards, allowed for the analysis of available AR technology in various settings and types of augmented reality systems. Overall, these articles provide an overview of AR technology used in varying degrees of success to influence dietary behaviors, portion size estimation, dietary knowledge, and physiological and psychological responses.

All participants were drawn from adult populations, including college students, pregnant women, and hospital patients. Several studies utilized quasi-experimental designs, but two cross-sectional, two experimental, and one mixed-method study were also included. Most augmented reality systems were mobile AR devices, which are portable devices such as smartphones and tablets that allow users to experience AR applications in real-life settings [32]. Two studies used AR hardware or 3D models.

### 3.2. AR Technology, Device Brand, and Models

There were various augmented reality systems and tools used for dietary measurement. We identified four categories of AR technologies used in dietary monitoring. These include (1) smartphone-based AR applications, (2) marker-based AR systems, (3) 2D/3D model-based AR systems, and (4) camera-based and external device systems.

### 3.3. Smartphone-Based AR Applications

Smartphone-based AR applications allow for the overlay of computer-generated content onto a user’s real-world views through the camera of a smartphone [33]. AR apps deliver real-time visual feedback regarding food intake. In this review, authors most often utilized smartphone-based AR applications. ChanLin (2019) [17] employed a smartphone-based AR nutrition monitoring system, which allowed participants to scan and analyze food images for macronutrients. Similarly, Saha et al. (2022) [18] integrated USDA’s MyPlate, a common tool used in public health nutritional research, with a smartphone app to enable food portion estimation through pre- and post-meal pictures. Foods such as fruits, dairy products, vegetables, grains, and proteins were assessed with the app to compare to food that was originally weighted. Alturki and Gay used an AR-based smartphone tool to conduct image recognition. The iPhone-based AR system was designed to instruct users about the nutrition content of Saudi-specific foods.

### 3.4. Marker-Based AR Systems

Several AR systems integrated fiducial markers or image recognition for portion size estimation. Fiducial marker systems are typically unique, physical patterns that computer vision algorithms can identify to find features in digital camera images. These systems are commonly used in AR technology [34]. For example, Mellos and Probst (2022) [5] integrated 3D food models from platforms such as TurboSquid and Sketchfab to aid a smartphone-based AR tool that included printed fiducial markers. Participants were tasked with estimating portion size for foods such as steak, broccoli, and rice. Brown and colleagues similarly designed ServARpreg to overlay portion sizes of carbohydrate-rich foods, such as rice, kidney beans, and pasta. In this study, pregnant participants overlayed portion sizes via smartphone cameras that were enabled with ZapWorks software.

### 3.5. 2D/3D Models

AR technology superimposes virtual 3D models in the real world to achieve an overlay between the virtual and real world [34,35]. 3D models are used for camera pose estimation and tracking. Edge-based tracking, which uses the edges of objects, can take the 3D model and project it into 2D. Another method is tracking camera motion, so the 2D displacement can help compute the 3D camera motion. Ho and colleagues conducted in Taiwan used an AR system that included 2D images and 3D models of Taiwanese foods, such as noodles and tempura, that were created through Agisoft Metashape and Sketchfab. Domhardt and colleagues similarly integrated a smartphone-based AR tool with physical markers to display 3D space for food shape tracing. Participants traced food images using the AR device to aid in estimating carbohydrates for meals such as meat patties, roast beef, and chicken.

### 3.6. Camera-Based and External Device Systems

External cameras and digital overlays were also employed by several authors. Rollo et al. (2017) [21] used an iPad Mini and Canon 5D Mark III camera to scan fiducial markers with the purpose of enhancing food measurement by participants. Study participants were tasked with matching real servings of food, such as vegetables, pasta, and beans, to AR servings of food.

Additionally, Pallavicini et al. (2016) [22] used AR technology to enhance cue exposure therapy, a process that involves reducing cravings and relapse by repeatedly showing individuals cues associated with their substance use in a controlled environment. These researchers used a Microsoft HD LifeCam to display AR-generated food alongside real food to determine if AR-based food could invoke a similar stimulus as real food.

Overall, all of the studies included in this review displayed the wide variety of AR systems that can be implemented in dietary monitoring. Each system varied in technology, device compatibility, and food measurement approaches to improve participant outcomes. However, there were differences in accuracy across the various AR studies.

### 3.7. Accuracy in Food Identification Variations

Accuracy outcomes varied considerably across studies and were strongly influenced by task type, food characteristics, AR system design, and user interaction. To clarify performance patterns, accuracy findings are presented separately for portion-size estimation and energy estimation, followed by synthesis across food categories.

Multiple studies demonstrated improvements when AR was used to support portion-size estimation. Mellos and Probst (2022) [5] reported a 12.2% improvement in estimation accuracy, while Ho et al. (2022) [19] found that 30.7% of AR-assisted estimates fell within ±10% of the true portion compared to lower accuracy in controls. Rollo et al. (2017) [21] similarly showed that AR overlays improved matching of served portions to standard reference sizes.

In contrast, energy estimation outcomes were more variable. Saha et al. (2022) [18] showed that although AR improved visual portion estimation for many foods, the AR system overestimated energy intake by 12.7%, particularly for amorphous or irregular foods. This distinction highlights that improvements in portion-size accuracy do not necessarily translate into accurate energy estimation.

Additionally, AR systems consistently improved accuracy for foods that were structured, such as broccoli, steak, and bread. This is likely due to clear boundaries and predictable shapes. Accuracy was significantly lower across studies for amorphous foods, including rice, mashed potatoes, and soups. Users had difficulty aligning overlays, tracing shapes, or interpreting volumes, which resulted in greater bias and variability.

Overall, AR technologies demonstrate promising benefits for visually structured items but still show limitations when applied to foods lacking a defined shape. These patterns point to the importance of developing improved AR modeling for amorphous foods and integrating multimodal sensing to enhance accuracy.

A summary of portion size and energy estimation metrics is available in Table 5.

### 3.8. Usability/Acceptability Assessment of AR Tool

The articles included in this review were analyzed for their usability or acceptability of the AR tool being tested. Table 6 details the definitions related to the usability and acceptability measures included in the review. Across studies, usability and acceptability of AR tools were generally rated positively. Standardized instruments such as the CSUQ, SUS, and user-satisfaction surveys showed that most participants found AR apps easy to use, visually engaging, and helpful for increasing awareness of portion sizes and nutrition knowledge. For example, Saha et al., 2022 [18] administered an adapted version of the User-Satisfaction Survey and the CSUQ to evaluate ease of use, satisfaction, adequacy of training, and information quality, with 80% of participants reporting high satisfaction with the AR system. High proportions of users (70–80%) reported satisfaction, ease of use, and perceived improvements in dietary awareness and healthy eating behaviors [18,19,20]. However, technical challenges such as marker alignment, overlay accuracy, and limited food options were noted as areas needing improvement [17,19,23].

Custom and qualitative evaluations provided similar insights. For example, Rollo et al., 2017 [21] designed a Likert-based usability questionnaire that assessed ease of aligning the device and food, confidence in using ServAR, clarity of overlay visuals, perceived helpfulness in everyday use, and the potential to improve healthy eating behaviors. Out of the study participants (N = 30), 80% either agreed or strongly agreed that the tool was easy to use and 73.3% agreed or strongly agreed that the tool aided participants in serving the appropriate serving size. Participants often favored 3D models over 2D for food estimation tasks and highlighted the value of AR tools in education and self-monitoring [19,25,26]. While real food remained easiest to quantify, digital models were still considered useful, especially with repeated exposure. Users appreciated interactive features like voice commands and tailored health information, noting that AR systems were motivating, accessible, and effective for raising dietary awareness [17,23]. Some limitations were identified, including time-consuming recording processes, but overall, AR technologies were considered acceptable, user-friendly, and supportive of healthier decision-making.

### 3.9. Effectiveness Outcomes

This systematic review also aimed to examine the effectiveness of augmented reality technology being applied as a dietary monitoring and control method compared to traditional methods. After review of the thirteen articles, four effectiveness outcomes were selected: portion size estimation, dietary knowledge, behavioral impact, physiological or psychological response. Table 6 includes the effectiveness outcomes and their respective definitions.

### 3.10. Portion Size Estimation

Portion size is the amount of food one chooses to eat at one time [37]. Portion size estimation involves “the ability to identify the weight and volume of a wide range of foods by visual observation through the conceptualization of food shapes” [5]. Four articles reported portion size estimation as a primary outcome when measuring the effectiveness of the AR tools as dietary control measures [5,18,19,21]. AR was explored as a tool to improve participants’ ability to estimate accurate portion sizes. Mellos and Probst found improvements (+12.2%) in estimation accuracy using smartphone-based AR, though results varied due to user familiarity, food type, and biases. Similarly, Saha et al. demonstrated that AR’s interactive 3D visualizations enhanced participants’ ability to compare perceived portions with standardized references, achieving a −6% error in portion size estimates, but still overestimating energy intake by 12.7%. AR technology increased awareness of portion sizes and improved accuracy from 19.4% to 42.9%, especially for non-amorphous foods, though performance remained inconsistent with items like noodles or soups [5]. AR estimates were also found to be more accurate than controls, with 30.7% of estimates falling within ± 10% of true portion size.

### 3.11. Dietary Knowledge

The second effectiveness outcome identified in this systematic review was dietary knowledge. This outcome highlights study efforts to improve participant dietary knowledge, such as the ability to identify foods and estimate macronutrients and portion sizes, as a method to measure dietary control. AR technology was able to significantly improve users’ ability to identify food groups and estimate nutrient content, particularly carbohydrates (Brown). Users showed greater accuracy in quantifying carbohydrate intake after using AR tools, with marker improvements from baseline to follow-up. However, improvements were less consistent when it came to identifying which foods contained carbohydrates, suggesting that while AR enhances quantification skills, it may be less effective for food recognition tasks. AR-based nutrition monitoring systems also demonstrated clear educational benefits (ChanLin et al., 2019 [17]). Users showed significant improvements in their understanding of nutritional concepts (*p* < 0.01), with measurable reductions in misconceptions and improved scores on knowledge assessments (*p* < 0.001). Post-test results consistently revealed enhanced learning outcomes, highlighting the potential of AR platforms as effective tools for nutrition education and dietary knowledge reinforcement.

### 3.12. Behavioral Impact

Two articles used their AR systems to measure the behavioral impact of the technology on user’s nutritional behavior [17,27]. The behavioral impact outcome is defined by AR’s technology’s ability to influence food choices, eating habits, and self-monitoring compliance. In this review, AR technology can support healthier decision-making by engaging users in interactive learning and self-control of food choices. Participants using AR tools were able track their intake, review diets over time, and adjust eating habits to align with nutritional guidelines. The interactive design increased cognitive engagement, reinforced learning through repeated scanning and reflection, and motivated some individuals to avoid less healthy options, such as fatty or oily foods. For example, AR has also been applied to influence behavioral and emotional responses to food. Studies demonstrated that virtual foods could elicit similar levels of palatability as real foods, particularly for high-calorie items, and that AR-based apps can effectively integrate personalized monitoring, dietary education, and behavior-change strategies. By combining features such as calorie tracking, portion guidance, healthy substitutions, and gamification, these tools encouraged fruit and vegetable consumption and supported weight-loss management. User feedback highlighted the value of AR’s accessibility and interactivity, showing promise for practical dietary control and long-term behavior change.

### 3.13. Physiological/Psychological Response

The final effectiveness outcomes were physiological or psychological responses. In this review, cue exposure therapy was a method used to stimulate physiological responses to real-life food. Cue exposure therapy (CET) is a treatment method based on Pavlov’s theory of conditioning, which involves individuals repeatedly exposes to cues associated with the substance or behavior without engaging in the substance itself. CET attempts to eliminate conditioned responses, like cravings for specific foods or habits [38]. The AR cue exposure therapy experiment allowed for the authors to determine if AR food stimulated the same emotional responses as real-life food (Pallavicini et al. (2016) [22]. The results showed a significantly higher arousal response after real food stimulus compared to photo stimulus and similar stimulus response to real food. Additionally, obese participants reported lower happiness levels after AR food exposure. Brown et al. (2019) [20] also assessed physiological and/or psychological responses to their ServARpreg system. During a process evaluation, 80% of participants agreed or strongly agreed that the AR device increased their awareness of how much they ate and increased their confidence in dietary knowledge. For example, one participant reported that the AR system “made me think more about the type and amount of food I was eating”. Additionally, some participants reported feeling anxious about measuring out the portion sizes despite aid from the AR system.

## 4. Discussion

This systematic review examined the usability and effectiveness of augmented reality technology as a method of dietary monitoring and control. Across the thirteen studies identified, four primary outcome domains were identified: portion size estimation, dietary knowledge, behavioral impact, and physiological/psychological response. Overall, AR interventions demonstrated promising benefits in enhancing dietary monitoring and control, with the most consistent improvements observed in portion size estimation accuracy and nutritional knowledge.

Across the thirteen studies, participants were predominantly adults, with several samples comprising college students (ChanLin et al., 2019 [17]; Ho et al., 2022 [19]), and others focusing on populations like pregnant women (Brown et al., 2019 [20]) and hospital patients (Domhardt et al., 2015 [24]). Most studies enrolled mainly young adults or individuals with reported low technological skills. Additionally, sample sizes were small and did not include participants who varied by race, socioeconomic status, or culture. Future studies should include larger and heterogeneous samples to garner a more comprehensive understanding of how AR tools perform across different subgroups, especially those with lower technological skills or higher risk of chronic disease. Furthermore, AR needs to be explored in real-world or clinical settings. The observed studies were limited to experimental or educational environments. Lastly, longitudinal studies can determine sustained engagement with AR tools and if initial improvements in dietary behaviors continue over time and impact long-term health outcomes.

Across the included studies, AR tools demonstrated promising improvements in portion size estimation and dietary monitoring; however, these enhancements were influenced by factors such as food type, user experience, and AR software. Several studies reported improved accuracy for structured foods like broccoli or steak, but decreased accuracy for amorphous foods such as rice, mashed potatoes, or soups. Saha et al. (2022) [18] exhibited reduced estimation error for several items but still overestimated caloric intake by 12.7%. This highlights that AR can produce errors depending on food characteristics.

Two studies demonstrated AR’s positive effect on nutrition knowledge. ChanLin et al. (2019) [17] reported that participants receiving high levels of AR engagement reported significant improvements in their understanding of nutritional concepts, along with increased adherence to app use. Brown et al. (2019) [20] similarly observed greater knowledge retention among pregnant women using the ServARpreg tool, particularly in carbohydrate quantification. These findings indicate that AR can be a valuable educational supplement, facilitating active learning and retention of dietary information through visual and interactive platforms. Three studies (ChanLin et al., 2019 [17]; Pallavicini et al., 2016 [22]; Alturki & Gay, 2019 [23]) reported that AR tools influenced users’ dietary behaviors. ChanLin et al. noted that students made healthier food choices and avoided high-fat foods after engaging with the AR system. Alturki and Gay incorporated behavior change techniques, such as gamification and goal tracking, within an AR app. This increased user motivation and dietary adherence. These findings suggest that AR technology can not only improve monitoring but also support habit formation through real-time feedback, personalization, and increased cognitive engagement. Two studies explored AR’s impact on emotional and physiological responses to food. Pallavicini et al. (2016) [22] demonstrated that AR food stimuli evoked similar arousal and emotional responses as real food, suggesting AR’s potential use in cue exposure therapy for binge eating or obesity treatment. Brown et al. (2019) [20] similarly reported that participants felt more aware of their eating habits when using the AR tool, though some experienced mild anxiety about accurately measuring food. These findings underscore AR’s ability to simulate real eating environments and influence emotional responses tied to dietary behaviors.

Usability was a major factor identified among the thirteen studies in this review that influenced the effectiveness and adoption of AR tools. Quantitative measures indicated that many participants found AR tools easy to use and helpful in guiding dietary behaviors. For example, over 70% of participants in studies by Saha et al. (2022) [18] and Brown et al. (2019) [20] reported high satisfaction and improved portion size awareness. Additionally, participants found the AR tools to be helpful in improving dietary choices. Qualitative feedback also reported the AR systems to be engaging and visually appealing. However, there were barriers reported by participants. Usability challenges persisted, particularly difficulties aligning fiducial markers or properly overlaying virtual models, were common and contributed to inconsistent performance. While AR might enhance the visual estimation process, its effectiveness depends heavily on interface design, visual clarity, and the user’s ability to manipulate the device. Overall, AR technology is a promising solution to improve existing dietary control methods, but improvements are needed in design, interface responsiveness, and training support, while also being accessible and adaptable for varying populations.

### 4.1. Future Improvements

As shown in this review, no studies measured AR use over time in longitudinal-based studies. There is potential to use AR for long-term use. AR tools have the capability to be developed for hands-free operation, such as continuous observation while eating or cooking, that integrates into individuals’ daily lives. Many AR systems also include real-time feedback with the ability to overlay portion size visuals or nutritional information during food selection or consumption to reinforce healthy food choices and self-monitoring [14,21]. AR tools can increase engagement and habit formation, especially through reminders and visual feedback. Additionally, AR tools have the potential to be integrated with current wearables and smart devices to monitor fitness and diet behaviors to give a more comprehensive view on one’s health. Finally, AI is underutilized among the AR tools in this review, and AI has the potential to learn user preferences and dietary needs to provide customized feedback to improve user satisfaction.

There are innovative AR devices in development to mitigate any issues present among the devices presented in this review. AR glasses were underutilized among the thirteen studies in this review. AR glasses are an innovative tool to integrate with existing dietary control methods due to the ability to combine several measurement tools. The AR systems discussed in this review lacked multimodal technology. AR glasses with multimodal sensors, including cameras, inertial sensors, and microphones, go beyond single modal sensors by measuring various processes of consuming food [39]. This includes measuring chewing and eating speed, swallowing counts, and visual observation of the foods being consumed. Multimodal systems can increase the accuracy of recorded food intake and give users a more realistic output of the foods consumed during use. The AR systems reported in this review could be enhanced with multiple sensors. For example, Pallavicini et al., 2016 [22] measured cue exposure with an external camera but including multiple body sensors could improve our understanding of how AR food versus real food elicited similar responses.

### 4.2. Limitations

AR tools can also be limited during long-term use. Many AR systems, such as AR glasses, are expensive and less available compared to smartphones or tablets. Depending on the design of the AR tool, some participants might feel self-conscious using or wearing it in public. There are also technological limitations to AR tools, such as battery life and visual clarity depending on the lighting and eyesight of the user. As displayed by the studies in this review, there is the possibility for user fatigue and challenges. Long-term use could lead to reduced adherence levels, especially if the AR glasses or app are hard to use or uncomfortable to wear.

Additionally, this review has several methodological limitations. First, the search was limited to PubMed, CINAHL, and Embase. Although these databases produced public health focused AR studies, relevant computing-focused work captured in Scopus and Web of Science may have been missed. Second, the review was not preregistered and did not follow a formal reporting protocol, which may limit reproducibility. A formal reporting protocol will be conducted retrospectively. Third, although all screening decisions were independently verified by a second reviewer, initial title, abstract, and full text screening were not conducted in fully blinded duplicate, and data extraction was not performed in full duplicate, which may introduce reviewer bias. Finally, substantial heterogeneity across study designs, populations, AR systems, and reported outcomes precluded quantitative synthesis, limiting our ability to assess effect sizes across studies.

## 5. Implications/Conclusions

This review highlights the growing potential of using AR technology to serve as an innovative tool for dietary monitoring and control, offering insights that extend beyond earlier reviews focused on food labeling, nutrition education, or general AR usability. Unlike previous work, this review integrates three core domains: (1) usability (2) accuracy outcomes (3) and behavioral or psychological effects, to present a more holistic understanding of how AR influences real-world dietary behaviors. By examining both monitoring and control processes, this review highlights AR’s capability not only to present information but to shape decision making, self-awareness, and eating patterns. These improvements could support the development of more accurate, real-time dietary assessment tools to reduce reliance on self-reporting methods and reduce bias. However, the existing evidence based remains limited by small sample sizes, heterogeneity in intervention designs, and variability in device accuracy and usability across populations.

In addition, the methodological limitations of this review necessitate cautious interpretation of these findings. Taken together, AR should be viewed as a promising but still developing approach rather than an established solution for dietary control.

Looking ahead, standardized AR development protocols, research in diverse populations, and studies conducted in real-world settings are needed to further evaluate the effectiveness more rigorously. Integration with artificial intelligence, multimodal wearable sensors, and real-time feedback systems represents a compelling direction for future innovation but should be considered as avenues for investigation rather than guaranteed enhancements. Strengthening collaboration between public health professionals, nutrition experts, and technology developers will be vital for creating AR tools that are both user friendly and capable of supporting dietary behavior change.

Taken together, this review provides a foundational map of the current evidence and clarifies the specific conditions under which AR technologies can enhance dietary monitoring and control. These insights can support the development of next-generation AR interventions and guide an evidence-driven research agenda for the field.

## Figures and Tables

**Figure 1 nutrients-17-03893-f001:**
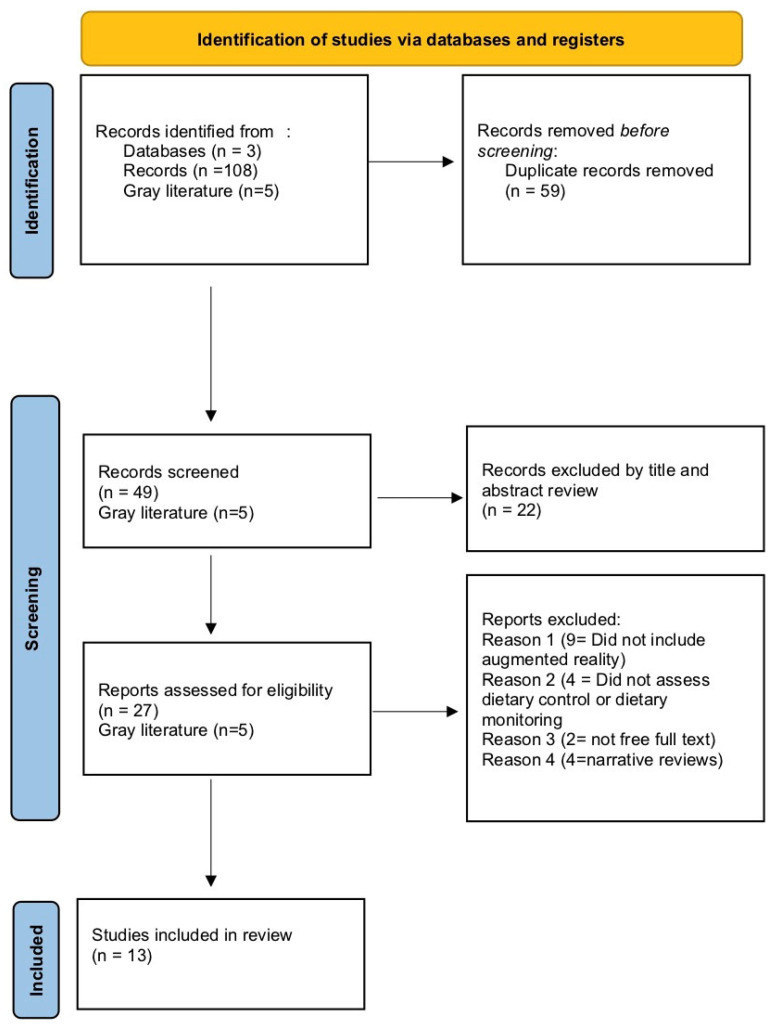
2020 PRISMA flow chart of study selection process.

**Table 1 nutrients-17-03893-t001:** Study design types.

Study Design	Definition
Experimental	Research method for investigating a cause-and-effect relationship between the predictor and outcome variables by manipulating an independent variable and observing its effect on the dependent variable
Quasi-experimental	Estimates the causal impact of an experiment by comparing groups, while lacking randomization found in experimental studies
Cross-sectional survey	Collecting a one-time survey from a sample at a single, specific timepoint to provide a snapshot of the relationship between variables
Mixed Method	Integrates both quantitative and qualitative research methods to gain deeper understanding of the relationship between variables

**Table 2 nutrients-17-03893-t002:** Risk of Bias Assessment of Randomized Controlled Trials.

	Randomization Process	Deviations from Intended Interventions	Missing Outcome Data	Measurement of the Outcome	Selection of the Reported Result	Overall
Brown et al. (2019) [20]	Some Concerns	Low Risk	Some Concerns	Low Risk	Low Risk	Some Concerns
Rollo et al. (2017) [21]	Low Risk	Low Risk	Low Risk	Some Concerns	Low Risk	Low Risk
Pallavicini et al. (2016) [22]	Low Risk	Low Risk	Some Concerns	Low Risk	Low Risk	Low Risk

**Table 3 nutrients-17-03893-t003:** Risk of Bias Assessment of Non-Randomized Studies.

	Bias Due to Confounding	Bias in Selection of Participants into the Study	Bias in Classification of Interventions	Bias Due to Deviations from Intended Interventions	Bias Due to Missing Data	Bias in Measurement of Outcomes	Bias in Selection of the Reported Result	Overall
ChanLin et al. (2019) [17]	Serious Risk	Moderate Risk	Low Risk	Low Risk	Low Risk	Moderate Risk	Low Risk	Serious Risk
Saha et al. (2022) [18]	Moderate Risk	Moderate Risk	Low Risk	Low Risk	Low Risk	Moderate Risk	Low Risk	Moderate Risk
Ngan Ho et al. (2022) [19]	Serious Risk	Moderate Risk	Low Risk	Low Risk	Low Risk	Moderate Risk	Low Risk	Serious Risk
Mellos and Probst (2022) [5]	Moderate Risk	Moderate Risk	Low Risk	Low Risk	Low Risk	Moderate Risk	Low Risk	Moderate Risk
Alturki and Gay (2019) [23]	Serious Risk	Moderate Risk	Low Risk	Low Risk	Low Risk	Serious Risk	Low Risk	Serious Risk
Domhardt et al. (2015) [24]	Serious Risk	Moderate Risk	Low Risk	Low Risk	Moderate Risk	Moderate Risk	Low Risk	Serious Risk
Stutz et al. (2014) [25]	Moderate risk	Moderate Risk	Low Risk	Low Risk	Low Risk	Low Risk	Low Risk	Moderate Risk
Chun Lam et al. (2021) [26]	Moderate Risk	Moderate Risk	Low Risk	Low Risk	Low Risk	Moderate Risk	Low Risk	Moderate Risk
Narumi et al. (2012) [27]	Low Risk	Moderate Risk	Low Risk	Low Risk	Low Risk	Moderate Risk	Low Risk	Moderate Risk
Dinic and Stutz (2017) [28]	Moderate Risk	Moderate Risk	Low Risk	Low Risk	Low Risk	Moderate Risk	Low Risk	Moderate Risk

**Table 4 nutrients-17-03893-t004:** Systematic review matrix detailing study results and primary outcomes.

Author/Year	Study Population and Sample Size (n)	Study Design	Augmented Reality System	Device Brand/Model	Usability	Monitoring Effectiveness (Food Identification and Portion Size Estimation)	Control Effectiveness (Dietary Knowledge, Behavioral Impact, Physiological/Psychological Response)
ChanLin, 2019 [17]	College Students N = 65	Mixed Methods 1 Arm: pre post test	Mobile AR Device	Samsung Smartphone	Feasibility and Acceptability Qualitative Interviews (100% improved dietary awareness)	NR	Dietary Knowledge Behavioral Impact
Saha, 2022 [18]	Adults N = 15	Feasibility study/cross-sectional 1 Arm: pre post test	Mobile AR Device	Smartphone App (USDA MyPlate)	Feasibility & Acceptability Adapted User Satisfaction Survey: 80% satisfied/extremely satisfied with app Computer System Usability Questionnaire: Mean CSUQ score 35.3 (13.5)	Portion Size Estimation Food Identification	NR
Ngan Ho, 2022 [19]	College Students N = 65	Quasi-experimental education intervention study 1 Arm: pre post test	3D and 2D dietetic training models	Agisoft Metashape Professional Addition	Feasibility and Acceptability Custom Questionnaire (receptiveness and student perceptions) 71% found digital food models helpful in conducting virtual dietary assessments; 82% supported educational integration of digital food models into Nutritional Practicum course	Portion Size Estimation Food Identification	NR
Mellos and Probst, 2022 [5]	Adults N = 26	Quasi-experimental Study 3 Arms: Control, Online, AR	Mobile AR Device	App: Javascriptlibraries 3D food images: turbosquid, sketchfab, cgtrader Food models and Fiducial marker: Blender software Portion Size: FoodWorks	NR	Portion Size Estimation Food Identification	NR
Brown, 2019 [20]	Pregnant Women N = 40	1 Arm: pre post test RCT	Mobile AR Device	AR Platform: Zapworks	Feasibility System Usability Scale (SUS) No total score reported; Findings suggest above average usability	Portion Size Estimation Food Identification	Dietary Knowledge
Rollo, 2017 [21]	Adults N = 90	User study (randomized experimental design) 3 Arms: Control, verbal, ServAR RCT	Mobile AR Device	AR Platform: Zapworks	Custom Questionnaire; mean scores between 1.7 and 2.1, indicating easy to use)	Portion Size Estimation	NR
Pallavicini, 2016 [22]	Adults N = 30	Experimental Exploratory Study 2 Arm: Control and Experimental RCT	External camera and AR plate marker	Microsoft’s HD LifeCame Marker for AR stimulus	Not reported	NR	Physiological/ Psychological Response
Alturki and Gay, 2019 [23]	Adults N = 26	Qualitative pilot study 1 Arm	Mobile AR Device	iPhone camera and app	Feasibility & Acceptability Expert Usability Testing and Qualitative Interviews (100% found app interactive and easy to navigate)	Food Identification	Dietary Knowledge
Domhardt et al., 2015 [24]	Hospital Patients N = 6	Quasiexperimental Study Multi-Method Analysis 1 Arm: pre post test	Mobile AR Device	USDA National Nutrient Database for Standard Reference (USDA-NNDSR)	Feasibility Qualitative Methods (verbalizing concerns while using tech; researcher observation; follow-up interviews) (100% agreed difficulty aligning AR markers, using interface and user frustration)	Portion Size Estimation Food identification	NR
Stutz et al., 2014 [25]	Students N = 28	User study (comparison group design) 4 Arm: visual estimation only; estimation using own fist size; mobile AR with 3-point input; mobile AR with mesh deformation input	Mobile AR Device	Mobile AR with 3-point input and mesh deformation; fiducial marker	Acceptability Qualitative Open-ended Questions 3-point interface preferred; mesh error prone	Portion Size Estimation	
Chun Lam et al., 2021 [26]	Adults N = 36	Usability and evaluation study	Mobile AR Device	3D food models using photogrammetry and rendering models Asus ZenFone AR	Feasibility & Acceptability System Usability Scale (SUS); SUS = 76, “Good usability/accepted by users”	Portion Size Estimation	NR
Narumi et al., 2012 [27]	Adults N = 12	Experimental study Repeated measures 3 arms, all participant completed all arms	Custom-built headset system	Head-mounted video see-through display system that scales apparent size of food	NR	NR	Physiological/ Psychological Response Behavioral Impact
Dinic and Stutz, 2017 [28]	Adults N = 16	Repeated measures One arm Compare user-estimated food volume vs. actual measured volume	Mobile AR Device	Android phone with built-in depth sensor; 3D mesh	Acceptability System Usability Scale (SUS) reported in previous study	Portion Size Estimation	NR

Many studies did not report standardized accuracy metrics such as mean absolute percentage error (MAPE), Bland–Altman limits of agreement, or systematic bias estimates. Values are presented as reported in the original publications. AR = augmented reality; NR = not reported; RCT = randomized controlled trial; USDA = U.S. Department of Agriculture.

**Table 5 nutrients-17-03893-t005:** Summary of Portion Size Estimation and Energy Estimation Metrics Across Studies.

Study	Task	Ground Truth Reference	Error Metric	Direction of Bias	Food Category
Saha et al. 2022 [18]	Portion size and energy estimation	Weight back scale method	Mean % error	Overestimation of energy intake by 12.7%; improved portion size accuracy	Amorphous & structured
Mellos and Probst 2022 [5]	Portion size estimation	FoodWorks reference volumes	% improvement in accuracy	+12.2% Improvement	Structured solids
Ho et al., 2022 [19]	Portion size estimation using 2D/3D AR	Food models calibrated to true weights	% within ± 10% of true	Improved accuracy for structured foods; poorer for noodles/soups	Mostly amorphous
Rollo et al., 2017 [21]	Portion matching	Weighed food items	Correct match rate	Higher accuracy in AR arms than controls	Structured
Domhardt et al., 2015 [24]	Carbohydrate estimation	USDA NNDSR reference	Accuracy score	Underestimation for amorphous foods; better structured items	Mixed

**Table 6 nutrients-17-03893-t006:** Effectiveness outcomes, measures, and definitions.

Effectiveness	Measures	Definition
Monitoring Effectiveness	Portion Size Estimation	Did the AR tool improve portion estimation compared to traditional methods? (e.g., *percentage improvement in estimation accuracy, reduction in error rates*).
	Food Identification	Did the AR tool correctly recognize and distinguish different types of foods?
Control Effectiveness	Dietary Knowledge	Did AR improve users’ ability to identify foods, estimate macronutrients, or understand dietary guidelines?
	Behavioral Impact	Did AR influence **food choices, eating habits, or self-monitoring compliance**? (e.g., *did users continue logging intake over time?*).
	Physiological or Psychological Response	How did AR influence **hunger, cravings, or emotional responses** to food?
Usability	Feasibility	**Feasibility** is the extent to which an intervention or system can be successfully used or implemented in a specific setting, considering practical constraints like time, resources, and user capacity.
	Acceptability	**Acceptability** refers to the extent to which target users perceive an intervention or tool as appropriate, engaging, and agreeable based on their cognitive and emotional responses. (Sekhon et al., 2017 [36])

## Data Availability

No new data were created or analyzed in this study. Data sharing is not applicable to this article.

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
