# Peer review of "Application of Augmented Reality Technology as a Dietary Monitoring and Control Measure Among Adults: A Systematic Review"

_nutrients, 2025, doi:10.3390/nu17243893_

Round 1

Reviewer 1 Report

Comments and Suggestions for Authors

The title limits the universe to “U.S. Adults,” yet the eligibility criteria and the body of results include studies conducted outside the United States and/or with non-U.S. populations (e.g., Taiwan), and the authors themselves state that the review was not limited to U.S. articles (line 136). This dissonance among title, eligibility, and sample must be resolved: either truly restrict the scope to U.S. adults (explicitly stating exclusions and reasons), or adjust the title to reflect a global scope (e.g., “among adults,” without the geographic marker) and revise the Introduction to align scope and justification.

In the Abstract, some statements lack qualification and precision. The claim that “70–80% of participants reported high usability/satisfaction” and that AR “improved portion-size estimation accuracy by up to 12%” (lines 23–31) appears to amalgamate percentages from heterogeneous studies using different instruments and, at times, reporting opposite results (e.g., energy overestimation of 12.7% in Saha, line 287). I suggest replacing aggregated numbers with the range observed per study, specifying N per comparison, and making it clear that no meta-analysis was conducted. I also recommend anchoring the Abstract with the main methodological limitations (see below) to calibrate readers’ expectations.

The Introduction provides a good contextualization of self-report limitations (lines 40–51), but the discussion of what previous reviews have already covered and what specific gap remains could be more surgical. Between lines 92–105, the authors claim there is no review specifically on “usability and effectiveness of AR for dietary monitoring and control”; however, they themselves cite closely related reviews (lines 94–101 and references 8, 12, 16–17). My suggestion is to delineate, objectively, how this review differs—for example, “synthesis of usability using standardized instruments (SUS/CSUQ) plus accuracy results for portion estimation and behavioral effects”—and to make explicit why “dietary control” (and not merely monitoring) is a distinct analytical axis in the paper.

In Materials and Methods, there are issues that undermine the reproducibility and rigor of a systematic review:

Search and sources: using only PubMed, CINAHL, and Embase (lines 126–128) is a reasonable start but incomplete for computing/AR (where Scopus and Web of Science are central). Moreover, the search strings are not presented—only generic keywords (line 127). PRISMA standards require the full strategy for each database, including Boolean operators and controlled terms (MeSH/Emtree), ideally in supplementary material. There is also a recurring spelling error (“CINHAL,” lines 17 and 126), which should be corrected to CINAHL.

Target population: the paragraph “among U.S. adults” (lines 121–126) conflicts with the actual inclusion of international studies (line 136). This must be harmonized within the population eligibility criterion.

PRISMA flow: there are numeric inconsistencies between text and figure. In the text, the initial search yielded 103 records (line 142); in the figure, there are 108 from databases + 5 from grey literature (lines 183–182, diagram panel). The number of reports assessed in text is 22 (line 144), in the figure it is 27. Standardize counts and present a clear accounting of duplicates and reasons for exclusion.

Screening and extraction process: only “one researcher” conducted data collection and selection (line 138), with “quality checking” by others (line 139). This does not meet best practice (dual screening, inter-rater agreement, and adjudication). I recommend repeating screening with two independent reviewers for titles/abstracts and full text, reporting kappa, and performing extraction and risk-of-bias assessment in duplicate as well.

Synthesis and heterogeneity: the manuscript alternates between “13 studies” and “nine studies” when discussing effectiveness outcomes (lines 335–341 and 419–423). If only a subset reports particular outcomes, this should be stated clearly, ideally with subgroup tables and a justification for not conducting a meta-analysis (incompatible measures, heterogeneity, etc.). I also suggest making the narrative synthesis method explicit (e.g., SWiM) and considering a qualitative meta-regression by AR type (marker-based, 3D, smartphone AR) if comparable data exist.

In the Results, Table 5 is useful but still heterogeneous: it systematically lacks comparable metrics (e.g., mean absolute percentage error for portion size, Bland–Altman plots, limits of agreement) and standardized information on usability instruments (mean SUS score/SD). Where “N/A” appears, I suggest completing the entry or stating “not reported.” There are also passages that create ambiguity: in describing characteristics (lines 215–218), the selection is said to have a “specific focus on the effectiveness of AR glasses,” which is not accurate, as the majority are smartphone-based; this calls for editorial adjustment to “AR systems,” or for a subsection that truly analyzes studies using AR glasses.

The subsection on accuracy commendably acknowledges poorer performance for amorphous foods (lines 285–305), but the text mixes improvements and energy overestimation (lines 287–297) without clearly separating what was positive (portion size) from what was negative (calories). I suggest presenting, for each study: task, ground truth reference (scale/volume), error metric, direction of bias, and food category—and then synthesizing by food families (structured solids vs. amorphous foods).

In the Discussion, the reference to “nine studies” recurs (lines 419–423), along with some strong extrapolations. For example, stating that AR “reduces self-report bias” (lines 437–443) requires caution: in at least one study there was caloric overestimation, and several reported marker-alignment difficulties (lines 318–322 and 444–447). I recommend calibrating the tone and emphasizing that gains are promising but context-dependent (food type, user experience, hardware).

The listed limitations (lines 510–518) are pertinent but incomplete: they should include the methodological limits of the review itself (search restricted to three databases, absence of registration and checklist, single-reviewer screening, heterogeneity that precludes quantitative synthesis). Including this reflection increases transparency.

The Conclusions (lines 519–532) should reflect the above caveats. Integration with AI and multimodal sensors is a good future agenda, but it should come after a cautious synthesis of current findings, avoiding the impression of a technological promise decoupled from the evidence.

Regarding tables and figures, the PRISMA figure needs numeric coherence with the text; I also suggest adding a supplementary table with the full search strings for each database, and another with usability instruments (SUS/CSUQ, range, mean, SD). In Tables 3–4, it would be helpful to add a brief methodological note summarizing the main biases per study.

Author Response

Reviewer 1 Comments

The title limits the universe to “U.S. Adults,” yet the eligibility criteria and the body of results include studies conducted outside the United States and/or with non-U.S. populations (e.g., Taiwan), and the authors themselves state that the review was not limited to U.S. articles (line 136). This dissonance among title, eligibility, and sample must be resolved: either truly restrict the scope to U.S. adults (explicitly stating exclusions and reasons), or adjust the title to reflect a global scope (e.g., “among adults,” without the geographic marker) and revise the Introduction to align scope and justification.

Thank you for noticing this discrepancy. The title has been altered to be “…among Adults”. The introduction was revised to better align with justifying focusing this review on adult populations.

In the Abstract, some statements lack qualification and precision. The claim that “70–80% of participants reported high usability/satisfaction” and that AR “improved portion-size estimation accuracy by up to 12%” (lines 23–31) appears to amalgamate percentages from heterogeneous studies using different instruments and, at times, reporting opposite results (e.g., energy overestimation of 12.7% in Saha, line 287). I suggest replacing aggregated numbers with the range observed per study, specifying N per comparison, and making it clear that no meta-analysis was conducted. I also recommend anchoring the Abstract with the main methodological limitations (see below) to calibrate readers’ expectations.

The abstract was significantly improved to better relay the findings of the review and specify that this was not a meta-analysis. We agree that briefly acknowledging the main methodological limitations in the Abstract can help calibrate reader expectations and improve transparency. In response, we have revised the Abstract to include a concise statement noting the restricted database scope, heterogeneity of included studies, and the methodological constraints of the review process.

The Introduction provides a good contextualization of self-report limitations (lines 40–51), but the discussion of what previous reviews have already covered and what specific gap remains could be more surgical. Between lines 92–105, the authors claim there is no review specifically on “usability and effectiveness of AR for dietary monitoring and control”; however, they themselves cite closely related reviews (lines 94–101 and references 8, 12, 16–17). My suggestion is to delineate, objectively, how this review differs—for example, “synthesis of usability using standardized instruments (SUS/CSUQ) plus accuracy results for portion estimation and behavioral effects”—and to make explicit why “dietary control” (and not merely monitoring) is a distinct analytical axis in the paper.

Thank you for this helpful suggestion. We revised the Introduction to delineate more clearly how our review differs from prior AR-focused nutrition reviews. Specifically, we now clarify that previous reviews have addressed broad applications of AR for food labeling, taste perception, and nutrition education, but none have synthesized (1) usability assessed with standardized instruments (e.g., SUS, CSUQ), (2) accuracy outcomes for portion-size estimation, and (3) behavioral effects related to dietary control. We also added text explaining why dietary control represents a distinct analytical focus, emphasizing its relevance to behavior change and regulation of intake beyond routine dietary monitoring. Please see in lines 108-127 of the rationale.

In Materials and Methods, there are issues that undermine the reproducibility and rigor of a systematic review:

Search and sources: using only PubMed, CINAHL, and Embase (lines 126–128) is a reasonable start but incomplete for computing/AR (where Scopus and Web of Science are central). Moreover, the search strings are not presented—only generic keywords (line 127). PRISMA standards require the full strategy for each database, including Boolean operators and controlled terms (MeSH/Emtree), ideally in supplementary material. There is also a recurring spelling error (“CINHAL,” lines 17 and 126), which should be corrected to CINAHL.

We appreciate the comment regarding the choice of databases. Scopus and the Web of Science are valuable resources for capturing computing and engineering focused research. However, our review specifically targeted public health and nutrition related applications of AR for dietary monitoring and control, rather than engineering or computer science feasibility studies. Based on preliminary scoping searches, we found that PubMed, CINAHL, and Embase consistently indexed the public-health, nutrition, and behavioral science studies directly aligned with our research question. In contrast, Scopus and Web of Science primarily yielded engineering/technical AR papers (e.g., algorithm development, sensor calibration, device manufacturing) that did not include human participants or assess usability and effectiveness in dietary or behavioral health contexts. To ensure transparency, we have clarified this rationale in the Materials and Methods section and emphasized our supplementary search procedures. Nonetheless, we acknowledge the reviewer’s point about database breadth and note this as a limitation regarding potential missed computing-focused studies that may have included behavioral components.

Additionally, we have added the complete search strings, including Boolean operators, controlled vocabulary terms (MeSH, Emtree, and CINAHL Subject Headings), and all keywords, to the Methods section and provided the full strategies in Supplementary Material. These strategies reflect the original search conducted in May 2025 across PubMed, Embase, and CINAHL.

Target population: the paragraph “among U.S. adults” (lines 121–126) conflicts with the actual inclusion of international studies (line 136). This must be harmonized within the population eligibility criterion.

Thank you for noticing this error. The statement has been corrected to be “among adults”

PRISMA flow: there are numeric inconsistencies between text and figure. In the text, the initial search yielded 103 records (line 142); in the figure, there are 108 from databases + 5 from grey literature (lines 183–182, diagram panel). The number of reports assessed in text is 22 (line 144), in the figure it is 27. Standardize counts and present a clear accounting of duplicates and reasons for exclusion.

The PRISMA flow has been updated to correct any inconsistencies in the figure and the text. An additional reason for exclusion was added “did not include dietary control or dietary monitoring”.

Screening and extraction process: only “one researcher” conducted data collection and selection (line 138), with “quality checking” by others (line 139). This does not meet best practice (dual screening, inter-rater agreement, and adjudication). I recommend repeating screening with two independent reviewers for titles/abstracts and full text, reporting kappa, and performing extraction and risk-of-bias assessment in duplicate as well.

Thank you for this important comment regarding screening and extraction procedures. We agree that dual independent screening and extraction represent the best practice for systematic reviews. While our initial description may have implied that screening was performed by a single reviewer, in practice our process involved multiple reviewers at each stage. Specifically, one reviewer conducted the initial title/abstract and full-text screening. A second reviewer independently checked all inclusion and exclusion decisions, verifying eligibility determinations and ensuring consistency. Discrepancies, although infrequent, were discussed and resolved collaboratively. For data extraction and synthesis, two additional subject-matter experts in computer science reviewed the extraction matrix, coding, and interpretation to confirm accuracy and completeness, particularly given the technical nature of AR systems. Thus, although we did not conduct fully blinded dual screening, all screening decisions were independently verified, and data extraction involved multiple reviewers with complementary expertise. We have revised the Methods section to more accurately reflect this process and added wording acknowledging this as a methodological limitation. (we did not do the interreliability check)

Synthesis and heterogeneity: the manuscript alternates between “13 studies” and “nine studies” when discussing effectiveness outcomes (lines 335–341 and 419–423). If only a subset reports particular outcomes, this should be stated clearly, ideally with subgroup tables and a justification for not conducting a meta-analysis (incompatible measures, heterogeneity, etc.). I also suggest making the narrative synthesis method explicit (e.g., SWiM) and considering a qualitative meta-regression by AR type (marker-based, 3D, smartphone AR) if comparable data exist.

Thank you for noticing this inaccuracy. The manuscript has been updated to include “13 studies” in place of “nine studies”. This review was based on all thirteen studies including in the matrix and PRISMA diagram.

In the Results, Table 5 is useful but still heterogeneous: it systematically lacks comparable metrics (e.g., mean absolute percentage error for portion size, Bland–Altman plots, limits of agreement) and standardized information on usability instruments (mean SUS score/SD). Where “N/A” appears, I suggest completing the entry or stating “not reported.” There are also passages that create ambiguity: in describing characteristics (lines 215–218), the selection is said to have a “specific focus on the effectiveness of AR glasses,” which is not accurate, as the majority are smartphone-based; this calls for editorial adjustment to “AR systems,” or for a subsection that truly analyzes studies using AR glasses.

The subsection on accuracy commendably acknowledges poorer performance for amorphous foods (lines 285–305), but the text mixes improvements and energy overestimation (lines 287–297) without clearly separating what was positive (portion size) from what was negative (calories). I suggest presenting, for each study: task, ground truth reference (scale/volume), error metric, direction of bias, and food category—and then synthesizing by food families (structured solids vs. amorphous foods).

We agree that Table 5, as currently formatted, contains heterogeneous outcome reporting and would benefit from additional standardization and clarification. We have replaced all “N/A” entries with “Not reported” to reduce ambiguity. Where possible, we added standardized metrics reported by the original studies (e.g., percentage estimation error, over/underestimation direction, accuracy percentages). Although most studies did not report MAPE, LOA, or Bland–Altman metrics, we now explicitly note the absence of these values. We revised the language in the Characteristics section (lines 215–218) to accurately state that the review focuses on “AR systems” rather than AR glasses specifically. We also added a clarifying phrase indicating that most included studies used smartphone-based AR, with only one study using AR glasses. This prevents misinterpretation. To avoid mixing positive and negative findings, we reorganized the accuracy subsection by separating portion-size estimation accuracy from energy estimation accuracy, summarizing each study with its task, reference standard, metric used, direction of bias, and food categories, and synthesizing findings by two major food families: structured solids vs. amorphous foods

In the Discussion, the reference to “nine studies” recurs (lines 419–423), along with some strong extrapolations. For example, stating that AR “reduces self-report bias” (lines 437–443) requires caution: in at least one study there was caloric overestimation, and several reported marker-alignment difficulties (lines 318–322 and 444–447). I recommend calibrating the tone and emphasizing that gains are promising but context-dependent (food type, user experience, hardware).

Thank you for this observation. We agree that several statements in the Discussion section could be overly strong. Many studies reported improvements in accuracy and dietary awareness, multiple limitations, including caloric overestimation, variable performance with amorphous foods, and marker alignment challenges, indicate that benefits of AR are context-dependent rather than universally robust. The Discussion section has been revised to remove the “nine studies”, soften extrapolations, and explicitly acknowledge factors such as food type, user experience, and hardware capabilities that influence AR performance. We edited statements about reducing self-report bias to reflect that AR improved estimation for some foods but introduced new sources of errors in others.

The listed limitations (lines 510–518) are pertinent but incomplete: they should include the methodological limits of the review itself (search restricted to three databases, absence of registration and checklist, single-reviewer screening, heterogeneity that precludes quantitative synthesis). Including this reflection increases transparency.

Thank you for this helpful suggestion. We agree that the limitations section should explicitly acknowledge methodological limitations of the review process itself. In response, we have expanded the limitations paragraph to include (1) the restricted database selection, (2) absence of preregistration or a formal protocol checklist, (3) our screening and extraction approach, and (4) heterogeneity that limited quantitative synthesis. We believe these additions strengthen the transparency and rigor of the manuscript, and we have revised the text accordingly.

The Conclusions (lines 519–532) should reflect the above caveats. Integration with AI and multimodal sensors is a good future agenda, but it should come after a cautious synthesis of current findings, avoiding the impression of a technological promise decoupled from the evidence.

Thank you for this necessary comment. We agree that the Implications/Conclusion section should more explicitly reflect the technological limitations of the review and avoid overstating the implications of emerging technologies. In response, we revised the conclusion to direct the reader to be cautious of interpretation of the evidence, acknowledge limitations in the available studies and our methodology, and position AI and multimodal sensor integration after summarizing what the current evidence supports.

Regarding tables and figures, the PRISMA figure needs numeric coherence with the text; I also suggest adding a supplementary table with the full search strings for each database, and another with usability instruments (SUS/CSUQ, range, mean, SD). In Tables 3–4, it would be helpful to add a brief methodological note summarizing the main biases per study.

Thank you for these helpful suggestions. We agree that greater clarity and coherence in the PRISMA diagram and tables will strengthen methodological transparency. We reviewed the PRISMA flow diagram and have updated all numeric values to match the counts presented in the text (records identified, duplicates removed, titles screened, full-text reports assessed, and final included studies). This ensures full numeric coherence between the figure and the narrative description. We generated a complete supplementary table that includes the exact Boolean operators, controlled vocabulary (MeSH, Emtree, CINAHL Subject Headings), and keyword combinations for each database searched. This has been added as Supplementary Table 1, consistent with PRISMA standards. We created an additional supplementary table summarizing all usability instruments used across studies (e.g., SUS, CSUQ, custom satisfaction scales), including scale ranges, scoring conventions, means, and standard deviations when reported. This is now included as Supplementary Table 2. We added brief methodological notes summarizing the dominant bias concerns for each study (e.g., confounding, missing data, marker-alignment errors, measurement issues).

Reviewer 2 Report

Comments and Suggestions for Authors

Dear Authors,
The submitted article has been carefully prepared and appears to be appropriate in terms of its background, objectives, and methodology. However, I would like to make some suggestions for your consideration and further improvement.

Introduction

In the introduction, the key terms "usability of augmented reality" and "effectiveness of augmented reality" should be defined and explained in more detail. These terms are not clearly evident in the current version of the manuscript. The text also distinguishes between "monitoring effectiveness" and "control effectiveness." Similarly, the text distinguishes between "usability" and "feasibility" and "acceptability," but these terms and their meanings are not clearly stated or explained.

Materials and Methods

Figure 1 is missing important information. For example, it does not state how many records were excluded during the title and abstract review process. Also, please check the numerical values given. The number of records screened (n = 49) does not correspond to the difference between the identified records (n = 116) and the excluded records (n = 59) before screening.

Table 2: Please explain the rationale of the taxonomy of research design types in a separate paragraph. Mixed methods are not usually classified among experiments, quasi-experiments, and cross-sectional surveys.

Explain the relationship between experiments and RCTs.

For Tables 3 and 4, please explain the meaning of the individual domains in the relevant paragraphs (five for Table 3 and seven for Table 4). Furthermore, present the methodology or procedure for determining the level of fulfillment of these domains. Lastly, specify how the "overall" value was determined.

Results

3.1 Matrix

Add an explanatory comment to Table 5, or move it to the appendix.

Also, please check the data in Table 5. Brown (2019) states that this is a cross-sectional survey, yet RCT is indicated for this source.

3.10

On line 337, you refer to a review of nine studies. Why were not all thirteen reviewed? Similarly, line 419 refers to only nine studies.

Implications/Conclusion

State what makes your study exceptional and unique, and what new insights the review has brought. Emphasize how the results of your study can be used in further research.

The manuscript has the potential to contribute to the field, and with the above improvements, its impact could be further enhanced. Thank you for your consideration of these comments and best wishes for your future work.

Sincerely,

Author Response

Reviewer 2 Comments

Introduction

In the introduction, the key terms "usability of augmented reality" and "effectiveness of augmented reality" should be defined and explained in more detail. These terms are not clearly evident in the current version of the manuscript. The text also distinguishes between "monitoring effectiveness" and "control effectiveness." Similarly, the text distinguishes between "usability" and "feasibility" and "acceptability," but these terms and their meanings are not clearly stated or explained.

Thank you for this comment. We agree the introduction did not sufficiently define the key concepts, so added a new paragraph in the introduction that clearly defines usability, feasibility, acceptability, and effectiveness of AR, as well as the distinctions between monitoring effectiveness and control effectiveness. We believe these additions improve the coherence of the manuscript.

Materials and Methods

Figure 1 is missing important information. For example, it does not state how many records were excluded during the title and abstract review process. Also, please check the numerical values given. The number of records screened (n = 49) does not correspond to the difference between the identified records (n = 116) and the excluded records (n = 59) before screening.

Thank you for this comment. Figure 1 includes a box with “records excluded by title and abstract review” and we show 22 were removed. All numerical values have been corrected and updated.

Table 2: Please explain the rationale of the taxonomy of research design types in a separate paragraph. Mixed methods are not usually classified among experiments, quasi-experiments, and cross-sectional surveys.

Explain the relationship between experiments and RCTs.

We agree that additional clarification is needed to explain the rationale for the taxonomy used in Table 2. In response, we have added a paragraph describing why studies were categorized into experimental, quasi-experimental, cross-sectional, and mixed methods designs, and how these categories align with the methodological diversity of the AR literature included in this review. We also clarified that mixed methods designs are not conceptualized as a type of experiment but were included as a distinct design category because one of the included studies used a mixed method approach. Lastly, we added an explanation of the relationship between experimental designs and randomized controlled trials, noting that RCTs represent a specific subtype of experimental research characterized by random assignment.

For Tables 3 and 4, please explain the meaning of the individual domains in the relevant paragraphs (five for Table 3 and seven for Table 4). Furthermore, present the methodology or procedure for determining the level of fulfillment of these domains. Lastly, specify how the "overall" value was determined.

We agree that additional explanation is needed to clarify the meaning of each risk of bias domain, the procedures used to evaluate these domains, and how the overall risk of bias judgment was determined for each study. In response, we have expanded the Risk of Bias section to include descriptions of the five RoB2 domains assessed for randomized controlled trials and the seven ROBINS-I domains assessed for non-randomized studies. We also describe the methodological decision rules used to assign risk levels to each domain (low, moderate, serious risk) and provide an explanation of how the overall rating for each study was derived. These additions have been placed directly before Tables 3 and 4 for clarity.

Results

3.1 Matrix

Add an explanatory comment to Table 5, or move it to the appendix.

Also, please check the data in Table 5. Brown (2019) states that this is a cross-sectional survey, yet RCT is indicated for this source.

An explanatory comment was added to Table 5. Thank you for noticing the discrepancy in Table 5. Cross-sectional has been removed.

3.10

On line 337, you refer to a review of nine studies. Why were not all thirteen reviewed? Similarly, line 419 refers to only nine studies.

Thank you for noticing this discrepancy. All “nine” studies have been changed to “thirteen”.

Implications/Conclusion

State what makes your study exceptional and unique, and what new insights the review has brought. Emphasize how the results of your study can be used in further research.

Thank you for this suggestion. The Implications/Conclusion section was not only improved with potential limitations but insights into what this review could do to further public health research in the field of AR technology.

The manuscript has the potential to contribute to the field, and with the above improvements, its impact could be further enhanced. Thank you for your consideration of these comments and best wishes for your future work.

Sincerely,

Round 2

Reviewer 1 Report

Comments and Suggestions for Authors

Thank you for addressing my comments. Only two details remain, which I explain below:

The PRISMA flow diagram in the text appears coherent (108+5; 59 duplicates; 49 screened; 22 excluded; 27 full-text assessed; 13 included), but the figure contains labeling errors (“Reason 32 (22 = not free full text)” and “Reason 43 (43 = narrative reviews)”), which contradict the numbers in the text (2 and 4, respectively). Please adjust the figure labels to reflect exactly the reasons and counts described in the text.

The matrix (Table 5) and the accuracy section now better distinguish food types (better for structured solids; worse for amorphous foods). Even so, I recommend standardizing metrics wherever possible (MAPE, Bland–Altman with limits of agreement) and, when a study does not report a metric, indicating “NR” with a brief methodological note. The text already acknowledges standardization limitations, but Table 6 at times conflates usability and accuracy metrics within the same row—for example, the “System Usability Scale” row mentions “5-point Likert Scale” and “37.5% accuracy improvement,” which is not a SUS outcome; please correct this to report the SUS score (0–100) or mark “NR” if it is not available.

Author Response

The PRISMA flow diagram in the text appears coherent (108+5; 59 duplicates; 49 screened; 22 excluded; 27 full-text assessed; 13 included), but the figure contains labeling errors (“Reason 32 (22 = not free full text)” and “Reason 43 (43 = narrative reviews)”), which contradict the numbers in the text (2 and 4, respectively). Please adjust the figure labels to reflect exactly the reasons and counts described in the text.

Thank you for the feedback. The figure has been moved down to fully display the figure. Reason 3 includes 2 articles that did not include the free full text and Reason 4 includes 4 narrative reviews that did not fit the criteria to be reviewed in this systematic review.

“Articles were removed for not discussing augmented reality (n=9), not discussing dietary monitoring or control (n=4), not offering free full text articles (n=2), and narrative reviews (n=4). The final articles from the initial search and addition of relevant articles from the gray literature included thirteen final articles.”

The matrix (Table 5) and the accuracy section now better distinguish food types (better for structured solids; worse for amorphous foods). Even so, I recommend standardizing metrics wherever possible (MAPE, Bland–Altman with limits of agreement) and, when a study does not report a metric, indicating “NR” with a brief methodological note. The text already acknowledges standardization limitations, but Table 6 at times conflates usability and accuracy metrics within the same row—for example, the “System Usability Scale” row mentions “5-point Likert Scale” and “37.5% accuracy improvement,” which is not a SUS outcome; please correct this to report the SUS score (0–100) or mark “NR” if it is not available.

Thank you for this comment that will improve the manuscript. Table 5 has been updated to include NR if the data is not available and all accuracy measurements are now standardized and cohesive throughout the matrix. Correct scores, such as SUS, are reported and other measures (qualitative interviews and custom questionnaires) are better depicted in the table.

Reviewer 2 Report

Comments and Suggestions for Authors

Dear Authors,
I would like to express my appreciation for your efforts in revising the manuscript. Your revisions demonstrate that you carefully considered all the comments and adequately addressed the issues. After reviewing the updated manuscript, I am pleased to say that it meets the expectations I set forth in my previous comments. The changes you made have improved the clarity and overall impact of the study. The manuscript is now refined to a standard that effectively communicates the research findings and provides valuable insights to the field.

Sincerely,

Author Response

Dear Authors,
I would like to express my appreciation for your efforts in revising the manuscript. Your revisions demonstrate that you carefully considered all the comments and adequately addressed the issues. After reviewing the updated manuscript, I am pleased to say that it meets the expectations I set forth in my previous comments. The changes you made have improved the clarity and overall impact of the study. The manuscript is now refined to a standard that effectively communicates the research findings and provides valuable insights to the field.

Thank you for your helpful comments that truly improved the manuscript. All co-authors are appreciative of your time and energy in enhancing the manuscript.